# Molecular Characteristics of Toxicity of Acrolein Produced from Spermine

**DOI:** 10.3390/biom13020298

**Published:** 2023-02-04

**Authors:** Keiko Kashiwagi, Kazuei Igarashi

**Affiliations:** 1Faculty of Pharmacy, Chiba Institute of Science, 15-8 Shiomi-cho, Chiba 288-0025, Japan; 2Amine Pharma Research Institute, Innovation Plaza at Chiba University, 1-8-15 Inohana, Chuo-ku, Chiba 260-0856, Japan

**Keywords:** acrolein, glutathione, oxidative stress, reactive oxygen species, spermine, spermine oxidase, tissue damage

## Abstract

Acrolein (CH_2_=CH-CHO), an unsaturated aldehyde produced from spermine, is one of the major contributors to oxidative stress. Acrolein has been found to be more toxic than reactive oxygen species (H_2_O_2_ and •OH), and it can be easily conjugated with proteins, bringing about changes in nature of the proteins. Acrolein is detoxified by glutathione in cells and was found to be mainly produced from spermine through isolating two cell lines of acrolein-resistant Neuro2a cells. The molecular characteristics of acrolein toxicity and tissue damage elicited by acrolein were investigated. It was found that glyceraldehyde-3-phosphate dehydrogenase (GAPDH); cytoskeleton proteins such as vimentin, actin, α- and β-tubulin proteins; and apolipoprotein B-100 (ApoB100) in LDL are strongly damaged by acrolein conjugation. In contrast, activities of matrix metalloproteinase-9 (MMP-9) and proheparanase (proHPSE) are enhanced, and antibody-recognizing abilities of immunoglobulins are modified by acrolein conjugation, resulting in aggravation of diseases. The functional changes of these proteins by acrolein have been elucidated at the molecular level. The findings confirmed that acrolein is the major contributor causing tissue damage in the elderly.

## 1. Introduction

Reactive oxygen species (ROS) such as superoxide anions (O_2_^-^), hydrogen peroxides (H_2_O_2_), and hydroxyl radicals (•OH), have been thought of as the main causes of tissue and cell damage in the elderly [1,2,3]. When spermine [NH_2_(CH_2_)_3_NH(CH_2_)_4_NH(CH_2_)_3_NH_2_], one of the polyamines which are essential for cell growth and viability [4,5], is metabolized by spermine oxidase, both acrolein (CH_2_=CH-CHO) and hydrogen peroxide (H_2_O_2_) are produced. The toxicity of acrolein and H_2_O_2_ was compared using a cell culture system, and acrolein was much more toxic than H_2_O_2_, a major compound of ROS, i.e., cell growth of mouse mammary carcinoma FM3A cells [6] was completely inhibited by 15 μM acrolein and 0.2 mM H_2_O_2_ [7,8].

Accordingly, it was examined whether acrolein is involved in the severity of brain infarction [9], dementia [10,11], renal failure [12], Sjögren’s syndrome [13], Parkinson’s disease [14,15,16], spinal cord injury [17], and diabetic nephropathy [18]. It was found that acrolein is strongly involved in the tissue damage of these diseases. During stroke, the level of the protein-conjugated acrolein (PC-Acro) in plasma increased, and the multiplied value of PC-Acro and polyamine oxidases (PAO; acrolein-producing enzymes consisting of spermine oxidase (SMOX) and acetylpolyamine oxidase (PAOX)) was nearly parallel with the size of brain infarction [9]. Production of acrolein in brain tissue was increased in aged mice due to an increase in spermine oxidase activity [19]. Furthermore, small brain infarction without obvious symptoms was identified with approximately 84% sensitivity and specificity by measuring PC-Acro together with interleuklin-6 (IL-6) and C-reactive protein (CRP) in plasma in clinical studies [20]. Through effective health care after measurement of these three biomarkers, the number of people with cerebral infarction gradually decreased during 7 years of evaluation [21]. Therefore, molecular mechanisms of cell and tissue damage by acrolein were investigated at the level of proteins because the SH group is most strongly impacted by acrolein [22,23]. In this review, we focused on the molecular characteristics of the toxicity of acrolein.

## 2. Characteristics of Two Cell Lines of Acrolein-Toxicity-Decreasing Neuro2a Cells

To examine how acrolein is detoxified in cells, mouse neuroblastoma Neuro2a cells were mutagenized with 0.1% ethylethanesulfonate and cultured in medium containing acrolein, which was gradually increased from 10 to 35 μM over 5 months. Two cell lines of acrolein-toxicity-decreasing, i.e., acrolein-resistant, Neuro2a cells were obtained (Figure 1) [24,25]. In the acrolein-toxicity-decreasing Neuro2a-1 (Neuro2a-ATD1) cells, the level of glutathione increased because two enzymes for glutathione synthesis (γ-glutamylcysteine ligase catalytic unit (GCLC) and glutathione synthetase GSHS)) were transcriptionally upregulated. Phosphorylation of c-Jun N-terminal kinase and that of c-Jun and NF-κB, which is involved in increased transcription, were both enhanced. It was confirmed that acrolein is detoxified by glutathione in these cells. The results strongly support the idea that one of the major functions of glutathione is the detoxication of acrolein, which is one of the major toxic compounds in cells.

It is thought that acrolein is produced from unsaturated fatty acids [26]. However, it was found that acrolein is mainly produced from spermine [27]. Acrolein is produced from spermine by two pathways. Firstly, spermine is oxidized to 3-aminopropanal, hydrogen peroxide, and spermidine by spermine oxidase (SMOX). Acrolein is readily produced from 3-aminopropanal non-enzymatically. Secondly, spermine is converted to *N*^1^-acetylspermine by spermidine/spermine *N*^1^-acetyltransferase (SAT1). Then, *N*^1^-acetylspermine is converted to 3-acetamidepropanal by acetylpolyamine oxidase (PAOX). Subsequently, acrolein is ineffectively produced from 3-acetamidepropanal non-enzymatically.

In the second cell line ATD2, both SMOX and PAOX decreased transcriptionally. Transcription factors FosB in AP-1 and C/EBPβ decreased in ATD2 cells, indicating that acrolein is mainly produced from spermine but not from unsaturated fatty acids.

## 3. Identification of Glyceraldehyde-3-phosphate Dehydrogenase (GAPDH) as an Acrolein-Conjugated Protein

We identified acrolein-conjugated proteins by gel electrophoresis in the S100 fraction of FM3A cells [6] treated with 40 μM acrolein for 9 h (Figure 2A). It was found that an approximately 37 kDa protein clearly decreased in acrolein-treated cells, suggesting that the acrolein-conjugated 37 kDa protein shifted to the P100 fraction or was hydrolyzed by proteases. The protein was identified as GAPDH by determining the peptide sequences by LC-MS/MS [28]. Amino acid residues conjugated with acrolein were determined after trypsin and endoproteinase Asp-N digestion and identified as Cys-150 at the active site and Cys-282 of GAPDH [29].

To determine whether acrolein-conjugated GAPDH is involved in cell damage, pcDNA-GAPDH was transfected into Neuro2a cells. In GAPDH-cDNA-transfected cells, the half maximal inhibitory concentration (IC_50_) of acrolein increased from 2.7 to 4.3 μM, indicating that inactivation of GAPDH by acrolein is strongly involved in cell growth [28].

It has been reported that nitric oxide (NO) reacts with Cys-150 of GAPDH and inactivates the enzyme, then inactivated GAPDH translocates to the nucleus with an E3 ubiquitin ligase Siah. In the nucleus, the complex activates P300/CBP acetylase, and GAPDH is acetylated. As a result, the cell undergoes apoptosis [30,31]. Therefore, it was determined whether this is also the case for acrolein-conjugated GAPDH. Although GAPDH in control cells existed mainly in the cytoplasm, GAPDH in cells treated with 8 μM acrolein for 6 h existed in both cytoplasm and nuclei (Figure 2B,C). Because nitrosylated GAPDH is acetylated [30,31], it was determined whether acrolein-conjugated GAPDH is also acetylated by immunoprecipitation with anti-acetyl-lysine followed by Western blotting with anti-GAPDH. The level of acetyllysine in GAPDH localized in nuclei increased in acrolein-treated cells. The results confirmed that acrolein-conjugated GAPDH translocates to nuclei and caused apoptosis. The percentages of TUNEL-positive cells were 1%, 39%, and 78% after treatment with 0, 4, and 8 μM acrolein, respectively, for 24 h (Figure 2D). It became clear that acrolein-conjugated GAPDH is acetylated and causes apoptosis similar to nitrosylated GAPDH [31]. Since GAPDH is an abundant and important enzyme in glycolysis, inactivation of GAPDH is thought to be one mechanism that underlies cell toxicity caused by acrolein. As for the acrolein conjugation at Cys-282, it was reported that the conjugation assists aggregation [32].

## 4. Acrolein Conjugation with Cytoskeleton Proteins

It was tested which protein can be conjugated with 10 μM acrolein in the P100 fraction of Neuro2a cells, and we found that a 57 kDa protein in P100 fraction is conjugated with acrolein, and this protein was identified as vimentin through the determination of amino acid sequences.

The cytoskeleton consists of 5–9 nm microfilament (actin), 8–12 nm intermediate filament (vimentin), and 25 nm microtubules (α- and β-tubulin proteins). Structural changes of three cytoskeletal proteins by acrolein were examined in Neuro2a cells (Figure 3A).

The acrolein-conjugated amino acid residue of vimentin was identified as Cys-328, which is the only cysteine in vimentin [33]. Time-dependent changes in cell shape of Neuro2a cells and distribution of vimentin in intermediate filaments were examined after addition of 10 μM acrolein. Dendritic spine extension was observed before addition of acrolein, but vimentin signal in dendritic spines decreased, and the projection became shorter after 3 h treatment of acrolein. The dendritic spines almost disappeared after 6 h treatment of acrolein, and cells shrunk after 24 h (Figure 3(Aa)).

It was also found that Cys-207, -257, and -285 and Lys-118 in actin were modified with acrolein. Changes of actin in Neuro2a cells cultured with 10 μM acrolein were also significantly similar to vimentin (Figure 3(Ab)). In addition, both vimentin and actin were degraded rapidly after acrolein conjugation [33].

Microtubules have been reported to be important for several aspects of normal brain function [34,35]. Therefore, we examined whether α- and β-tubulin proteins were also damaged by acrolein following brain infarction in the model mice with photochemically induced thrombosis (PIT). Since an increase in acrolein-conjugated α- and β-tubulin proteins was clearly observed, amino acid residues conjugated with acrolein were determined [36]. Those were Cys-25, -294, -347, and -376 in α-tubulin and Cys-12, -129, -211, -239, -303, and -354 in β-tubulin. Among them, two cysteine residues of α-tubulin (Cys-347 and -376) and four residues of β-tubulin (Cys-12, -129, -239, and -354) have been reported to exist at the interaction site of α- and β-tubulin proteins [37]. As a result, dendritic spine extension consisting of microtubules was greatly diminished in acrolein-treated Neuro2a cells (Figure 3(Ac)) and infarct brain tissue (Figure 3B).

**Figure 3 biomolecules-13-00298-f003:**
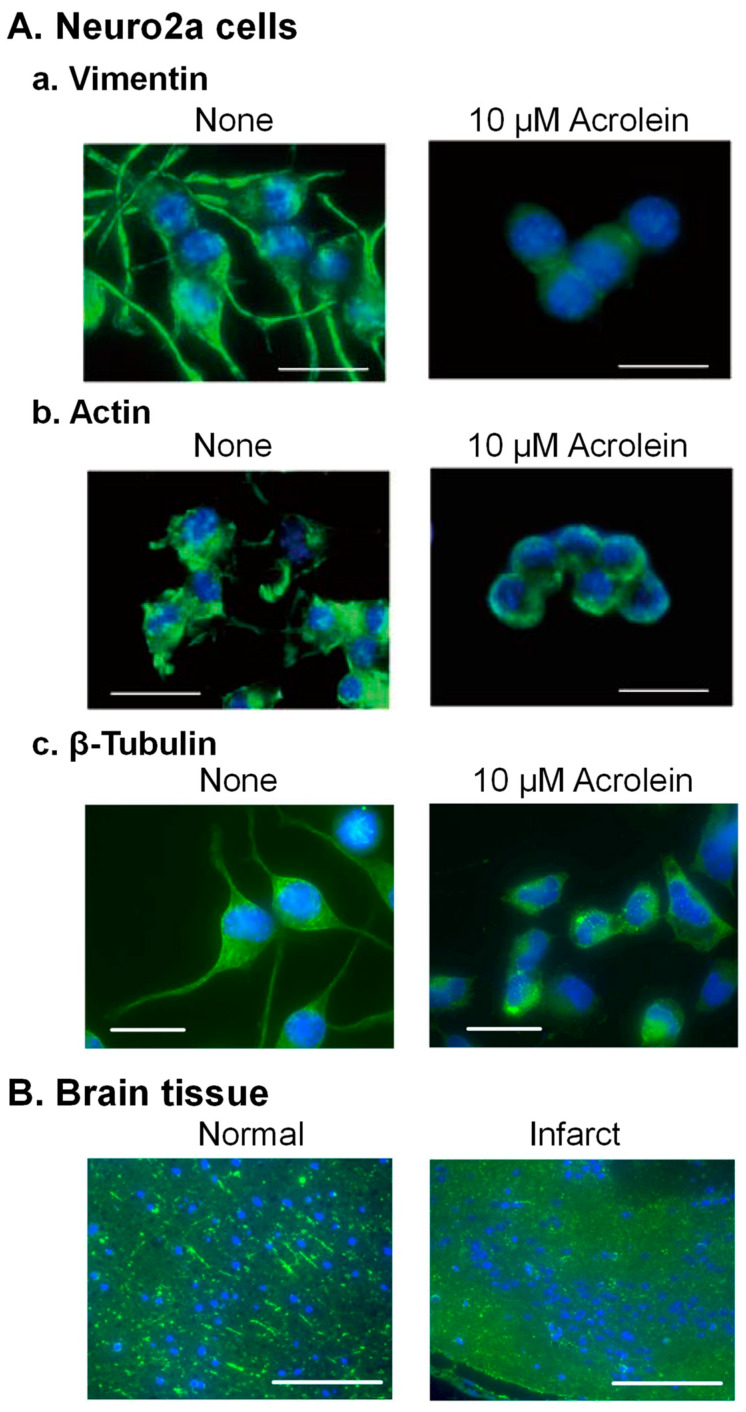
Change of the structure of cytoskeleton proteins through acrolein conjugation. (**A**). Neuro2a cells were cultured with 10 μM acrolein, fixed at 0 and 24 h, while intermediate (vimentin) filament was stained with anti-vimentin antibody (**a**), and microfilament (actin) was stained using Alexa Fluor 488 conjugated phalloidin (**b**). Microtubules (β-tubulin) were stained using anti-β-tubulin antibody (**c**). Bar = 10 μm. (**B**). A mouse brain was taken out at 24 h after induction of ischemia, and microtubules were stained using anti-β-tubulin antibody. Bar = 100 μm. Reprinted with permission from Ref. [33]. 2020 John Wiley and Sons and Ref. [36]. 2019 Elsevier.

These results indicate that acrolein causes functional defects in brain signaling through conjugations with the above three kinds of cytoskeleton proteins. The degree of the decrease in these proteins was in the order of vimentin > β-tubulin > actin in the brains of PIT model mice [33].

## 5. Acrolein Conjugation with Apolipoprotein B-100 (ApoB100) in Low-Density Lipoprotein (LDL)

Cerebral infarction is thought to be mainly caused by atherosclerosis, which initially arises from the foam-cell formation of macrophages [38]. For foam-cell formation, incorporation of oxidized low-density lipoprotein (LDL) into macrophages was thought to be the first event [39]. However, it was found that acrolein-conjugated LDL (Acro-LDL) is more readily taken up by macrophages, rather than oxidized LDL, through the SR-A1 scavenger receptor (Figure 4A) [40]. Foam-cell formation actually occurred through the uptake of Acro-LDL and brought about the accumulation of cholesteryl ester (CE) in lipid droplets.

Acrolein-conjugated amino acid residues in ApoB100 of LDL were identified by LC-MS/MS using Acro-LDL treated with 20 μM acrolein for 7 days. Among 4563 amino acid residues, 4061 residues of ApoB100 could be identified using peptides treated with trypsin and endoproteinase Asp-N. Among them, nine amino acid residues were acrolein-conjugated (Figure 4B,C). Four N-terminal amino acid residues (Cys-212 and Lys-327, -742, and -949) located at the SR-A1 recognition site were conjugated with acrolein (Figure 4C) [41]. It has been reported that negative charges are important for the binding reaction between LDL and macrophages [42]. Thus, acrolein conjugation with lysine residues is thought to decrease the positive charges and facilitate the interaction between Acro-LDL and SR-A1. The results indicate that acrolein can be a principal cause of atherosclerosis.

**Figure 4 biomolecules-13-00298-f004:**
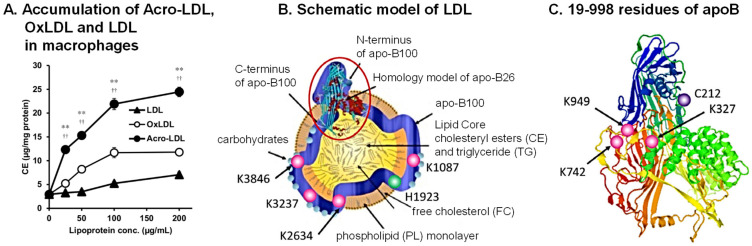
Preferential accumulation of acrolein-conjugated LDL in macrophages and identification of acrolein-conjugated amino acids in ApoB100. (**A**) Acro-LDL, OxLDL, and LDL were accumulated dose-dependently in THP-1 macrophages. ** *p* < 0.01 vs. OxLDL and ^††^
*p* < 0.01 vs. LDL. Schematic model of Acro-LDL particle (**B**) and 3D structure of peptide 19-998 of ApoB100 (**C**). Adapted with permission from Ref. [43]. 2009 Springer Nature. Acrolein-conjugated amino acids are shown. Adapted with permission from Refs. [40,41]. 2013 and 2021 Elsevier.

## 6. Activation of Proheparanase (proHPSE) through Acrolein Conjugation

Severe inflammation of acute ischemic stroke is brought about by the breakage of the blood–brain barrier (BBB) followed by the infiltration of monocytes and neutrophils into the brain [44,45]. Thus, it is effective to protect the BBB after the onset of ischemic stroke for the attenuation of post-ischemic inflammation. The endothelial glycocalyx, composed of membrane-bound glycoproteins and proteoglycans that wrap the lumen of endothelial cells, functions as a barrier against circulating cells. It consists of proteoglycans and sulfated glycosaminoglycans, including heparin sulfate and chondroitin sulfate. Hyaluronan, nonsulfate glycosaminoglycans that exists in a free form, are also major components of the endothelial glycocalyx [46]. The glycocalyx is degraded by heparanase (HPSE) [47,48]. HPSE (59 kDa) is matured from the precursor pre-proHPSE (68 kDa) via proHPSE (65 kDa). Thus, if proHPSE is activated, the endothelial glycocalyx becomes inactive as a barrier against circulating cells.

It was observed that levels of heparin sulfate and chondroitin sulfate decreased during stroke, and that activities of hyaluronidase 1 and HPSE increased in brain tissue of PIT model mice. Activity of HPSE in cerebral vessels increased after the onset of stroke (Figure 5A), and the volume of infarction was greatly diminished by co-administration of *N*-acetylcysteine (an acrolein scavenger) plus glycosaminoglycan oligosaccharides as compared with *N*-acetylcysteine administration alone [49].

**Figure 5 biomolecules-13-00298-f005:**
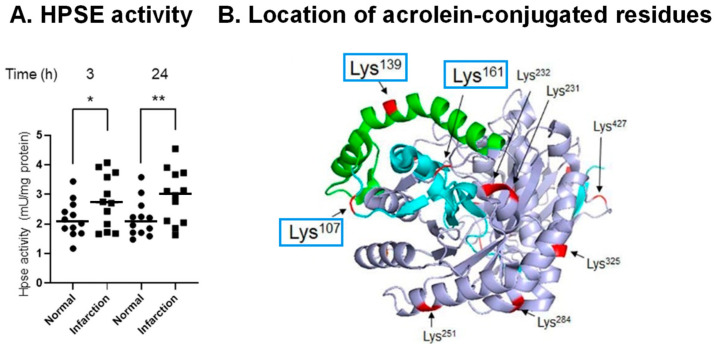
Acrolein-conjugated amino acids in proHPSE involved in the activation of HPSE. (**A**). Elevation of HPSE activity in infarct area compared to normal region at 3 and 24 h after onset of infarction in PIT model mice. *, *p* < 0.05; **, *p* < 0.01. (**B**). Location of acrolein-conjugated amino acid residues on proHPSE (PDB ID: 5LA4). Cyan, 8 kDa subunit; green, 6 kDa linker; light blue, 50 kDa subunit. Adapted from Ref. [49].

It was found that proHPSE is activated by acrolein as a result of acrolein conjugation with Lys-107, -139, and -161, which are located at the surface of proHPSE (Figure 5B). ProHPSE activity increased approximately 1.5-fold. Eleven other acrolein-conjugated lysine residues did not influence the proHPSE activity. Since proHPSE localizes outside cells through binding with heparan sulfate proteoglycans, the increase in the proHPSE activity through acrolein conjugation was strongly involved in the aggravation of brain stroke [49].

## 7. Activation Mechanism of Matrix Metalloproteinase-9 (MMP-9) by Acrolein in Saliva of Patients with Primary Sjögren’s Syndrome

Primary Sjögren’s syndrome (pSS) is a systemic autoimmune disorder mainly affecting the salivary and lacrimal glands to cause dry mouth and eyes as a result of reduced secretion from salivary and lacrimal grands because of destruction of these glands [50,51].

Matrix metalloproteinases (MMPs), particularly gelatinases (MMP-2 and MMP-9), have been reported to be involved in tissue damage of pSS patients [52,53,54]. Thus, it was examined whether the activity of MMP-9 derived from saliva or purified MMP-9 can be stimulated by acrolein. The protein level of 92 kDa MMP-9 in saliva of pSS patients was slightly higher (about 1.4-fold) than that in saliva of control subjects [55]. Activated forms of 82 and 68 kDa MMP-9 [56,57] were not detected in saliva in control subjects or pSS patients. The specific activity of MMP-9 in saliva of pSS patients was significantly higher (about 2.4-fold) than that in saliva of control subjects, consistent with the idea that MMP-9 in pSS patients is activated by acrolein. Indeed, concentration-dependent activation of MMP-9 by acrolein was confirmed using saliva of control subjects treated with 20–500 μM acrolein at 37 °C for 3 h.

Acrolein-conjugated amino acid residues were identified by LC-MS/MS using 92 kDa MMP-9. Residues of 11 cysteine, 2 lysine, and 2 histidine were conjugated with acrolein. Among them, it is thought that acrolein-conjugated Cys-99 in the propeptide domain is involved in the activation of MMP-9, similar to the activation by *S*-nitrosylation by nitric oxide (NO) at Cys-99 in MMP-7 [58]. A model of acrolein activation of MMP-9 is shown in Figure 6A. Similar to the cysteine switch in MMP-7, an interaction between Cys-99 and Zn^2+^ causing inactivation of MMP-9 is disturbed through acrolein conjugation with Cys-99, and Zn^2+^ can function as a co-activator of MMP-9 through interaction with the catalytic site. Although two His residues (His-405 and His-411) located at the active site were also conjugated with acrolein, the degree of acrolein conjugation was smaller than that with Cys-99. Accordingly, a significant inhibition of MMP-9 activity by acrolein conjugation with His-405 and His-411 was not observed. The effect of 50 μM acrolein and 100 μM histidine on purified MMP-9 activity is also shown in Figure 6B. Acrolein mainly increases the *k_cat_* value of MMP-9 activity.

**Figure 6 biomolecules-13-00298-f006:**
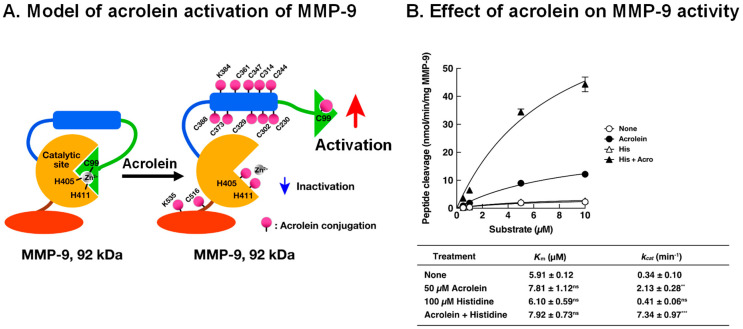
Activation model of MMP-9 by acrolein and effect of acrolein on enzymatic activity of MMP-9. (**A**). Activation model of MMP-9 by acrolein was created according to the cysteine switch model of MMP-7 [57]. Green, propeptide domain; blue, fibronectin repeats; orange, active site including Zn^2+^ binding site; brown, glycosyl domain; red, hemopexin domain. (**B**). The *K_m_* and *k_cat_* values of acrolein-activated MMP-9 with or without histidine. MMP-9 was incubated with 50 μM acrolein with or without 100 μM His at 37 °C for 3 h, and the MMP-9 activity was measured by changing the substrate concentration. The *K_m_* and *k_cat_* values were calculated using a Lineweaver–Burk plot. Adapted with permission from Ref. [55]. 2017 Elsevier. ** *p* < 0.01, *** *p* < 0.001.

## 8. Acrolein-Conjugated Immunoglobulin Increases Its Autoimmune Activity

It has been reported that autoantibodies against SSA (Ro) and SSB (La) proteins are often present in sera of pSS patients [50,51]. Thus, we investigated whether acrolein is involved in autoimmune activity. As shown in Figure 7A, the level of acrolein conjugation with immunoglobulins in saliva of pSS patients was elevated (>5-fold) compared to that in saliva of control subjects, although the immunoglobulin protein was only 1.5- to 2.0-fold higher in saliva of pSS patients [13].

The site of acrolein conjugation in immunoglobulin was then determined. Saliva of 11 pSS patients was collected, and immunoglobulins were purified. The amino acid residues conjugated with acrolein at the constant regions were λ (Lys-43), κ (Lys-75, Lys-80, His-81, Lys-82 and Cys-86), α-2 (Cys-300), γ-1 (Cys-27 and Lys-30), and γ-3 (Cys-297 and Lys-300). Twenty-four amino acid residues (20 cysteine and 4 lysine) conjugated with acrolein were also identified at the variable region of immunoglobulins (Figure 7B).

It was then examined whether acrolein treatment of saliva of control subjects increases the recognition ability for SSA (Ro) and SSB (La) proteins by incubating saliva with 25 and 50 μM acrolein for 48 h. As indicated in Figure 7C, acrolein significantly increased the autoimmune activity for SSA (Ro) and SSB (La). The recognition activity against both SSA (Ro) and SSB (La) proteins was increased approximately 1.5- to 2-fold by the treatment of saliva with 50 μM acrolein for 48 h, suggesting that acrolein modifies the recognition ability of immunoglobulins.

When amino acid residues at the variable region of immunoglobulins are conjugated with acrolein, the recognition ability of immunoglobulins can be changed to recognize a different antigen: i.e., proteins present in cells and tissues, such as SSA (Ro) and SSB (La).

## 9. Involvement of Acrolein during Brain Infarction and Dementia

This review outlines how acrolein causes tissue toxicity at the molecular level and is thus involved in diseases such as brain infarction and dementia. Photochemically induced thrombosis (PIT) model mice were prepared as described previously [59]. The volume of the infarction was determined by staining 2 mm-thick coronal slices with triphenyltetrazolium. This stains the viable brain tissue red, whereas infarct tissue remains unstained. Under our experimental conditions, the average volume of infarction at 24 h was 23 mm^3^. When *N*-acetylcysteine (250 mg/kg), a strong acrolein scavenger [22], was injected intraperitoneally, the average volume of infarction decreased from 23 to 16 mm^3^. PC-Acro at the locus of infarction greatly decreased, and polyamine content was increased significantly by the injection of *N*-acetylcysteine. Another acrolein scavenger, *N*-benzylhydroxylamine (200 mg/kg), also decreased the volume of infarction [27]. In addition, administration of *N*-benzylhydroxylamine decreased the volume of infarction 0 and 6 h after the onset of infarction, whereas edaravone (3-methyl-1-phenyl-2-pyrazolin-5-one), a scavenger of ROS, only decreased the volume of infarction administered at the onset of infarction. These results indicate that acrolein is strongly involved in the tissue damage during brain infarction.

There are reports that silent brain infarction (SBI) increases the risk of subsequent stroke [60]. It is, therefore, valuable to estimate SBI at the early period using biochemical markers. We found that measurement of PC-Acro together with IL-6 and CRP makes it possible to identify SBI with high sensitivity and specificity [20] and can decrease the number of people with cerebral infarction [21] as mentioned in the introduction.

Dementia, including Alzheimer’s disease (AD), is another serious disease affecting those at an advanced age, and its early detection is important for maintaining quality of life (QOL). Thus, we searched novel biomarkers for dementia. We first found that both protein-conjugated acrolein (PC-Acro) and Aβ_40/42_ in plasma increased in mild cognitive impairment (MCI) and Alzheimer’s disease (AD) patients [10]. However, these markers could not differentiate MCI patients from AD patients.

Next, biomarkers for dementia were searched in urine. It was found that amino acid (lysine)-conjugated acrolein (AC-Acro) and taurine in urine decreased in MCI and AD patients compared to control subjects, and the measurements of AC-Acro and taurine could differentiate MCI and AD patients. When AC-Acro and taurine were evaluated together with age using an artificial neural network model, median relative risk values for patients with AD and MCI and control subjects were 0.96, 0.53, and 0.06, respectively [61]. Since urine is relatively easy to collect, our findings provide a novel biomarker for dementia. This biomarker probably contributes to the maintenance of QOL of the elderly.

We have also reported that acrolein is detoxified by glutathione [24]. Acrolein is metabolized into 3-hydroxypropyl mercapturic acid (3-HPMA) after conjugation with glutathione, and 3-HPMA is excreted into urine. The level of 3-HPMA in urine after brain infarction also decreased significantly [62]. However, the measurement of 3-HPMA is not so easy. If a simple method to measure 3-HPMA is developed, the evaluation of dementia would become more accurate.

## 10. Concluding Remarks

Since acrolein is much more toxic than ROS, especially for proteins [22,27], molecular characteristics of acrolein toxicity were investigated. The results indicate that the activities of proteins including cysteine and lysine residues at the active site are regulated by acrolein, either negatively or positively. However, these acrolein-conjugated proteins always function negatively for cells and tissues (Figure 8). Since acrolein production from spermine increases in the elderly, it is important to clarify the molecular characteristics of acrolein toxicity to maintain good health. Although ROS are less toxic than acrolein, they are involved in DNA and RNA damage. Thus, it is also important to clarify ROS toxicity at the molecular level together with acrolein toxicity.

## Figures and Tables

**Figure 1 biomolecules-13-00298-f001:**
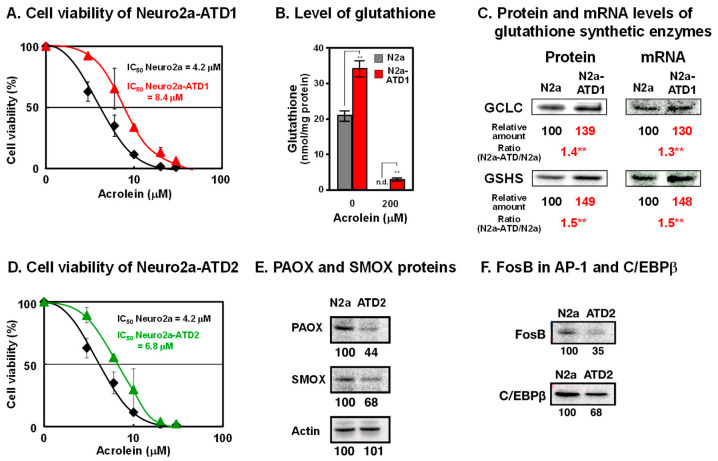
Characteristics of two cell lines of acrolein-toxicity-decreasing Neuro2a cells. (**A**). Cell viability of Neuro2a-ATD1. (**B**). Level of glutathione. (**C**). Protein and mRNA levels of glutathione synthetic enzymes. **, *p* < 0.01. (**D**). Cell viability of Neuro2a-ATD2. (**E**). Decrease in proteins of PAOX and SMOX in Neuro2a-ATD2. (**F**). Decrease in transcription factors involved in transcription of PAOX and SMOX in Neuro2a-ATD2. Adapted with permission from Refs. [24,25]. 2012 and 2016 Elsevier.

**Figure 2 biomolecules-13-00298-f002:**
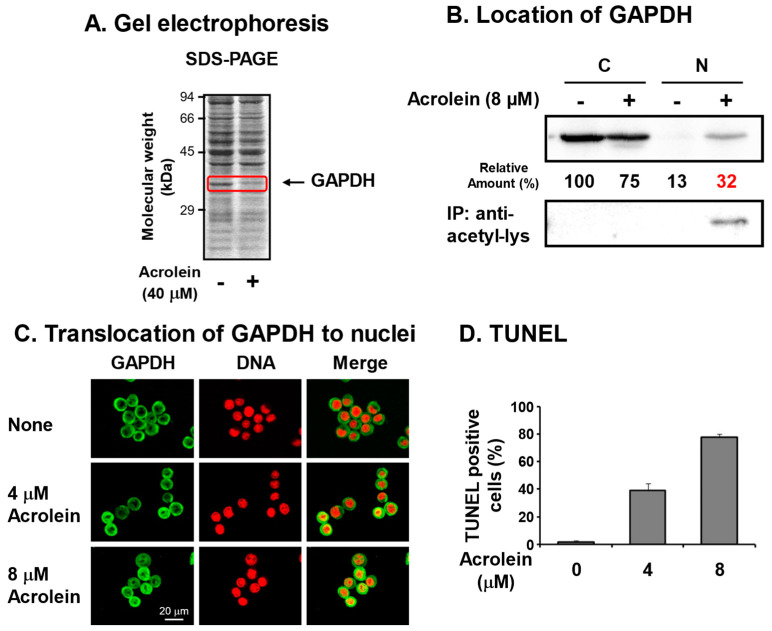
Identification of glyceraldehyde-3-phosphate dehydrogenase (GAPDH) as an acrolein-conjugated protein. (**A**). Identification of GAPDH as a reduced protein by acrolein treatment at 40 μM for 9 h in S100 fraction of FM3A cells by gel electrophoresis. (**B**). Nuclear localization of acrolein-conjugated GAPDH in an acetylated form. FM3A cells were treated with 0 or 8 μM acrolein for 6 h, and cytoplasm and nuclei were isolated. GAPDH and immunoprecipitated GAPDH by anti-acetyl-lysine in each fraction were identified by Western blotting. C, cytoplasmic fraction; N, nuclear fraction [28]. (**C**). Immunocytochemical detection of cells treated with 0, 4, and 8 μM acrolein for 6 h using GAPDH antibody and staining of DNA with propidium iodide. Images were merged using a confocal microscope. Bar indicates 20 μm. (**D**). The percentage of TUNEL-positive cells was indicated by counting approximately 500 cells. Values are means ± S. E. of triplicate determinations. Adapted with permission from Ref. [28]. 2013 Elsevier.

**Figure 7 biomolecules-13-00298-f007:**
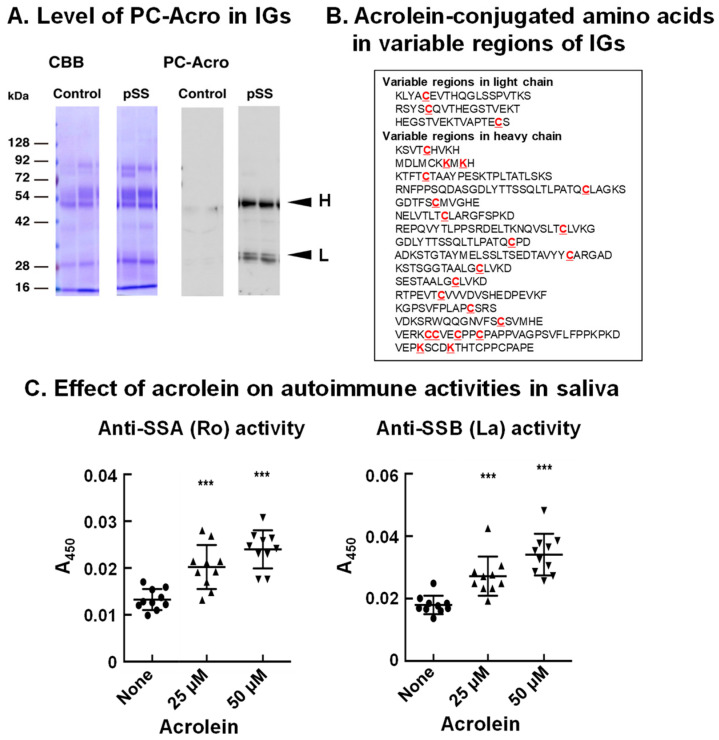
Increase in acrolein-conjugated immunoglobulins in pSS patients and determination of amino acids conjugated with acrolein in immunoglobulins. (**A**). Levels of acrolein-conjugated light and heavy chains of immunoglobulins in saliva from control and pSS subjects. (**B**). Acrolein-conjugated amino acids shown in red in variable regions of immunoglobulins in saliva from 11 pSS patients. (**C**). Effect of acrolein on recognition activities for Ro and La. Acrolein treatment at 37 °C for 48 h. ***, *p* < 0.001. Adapted with permission from Ref. [13]. 2015 Elsevier.

**Figure 8 biomolecules-13-00298-f008:**
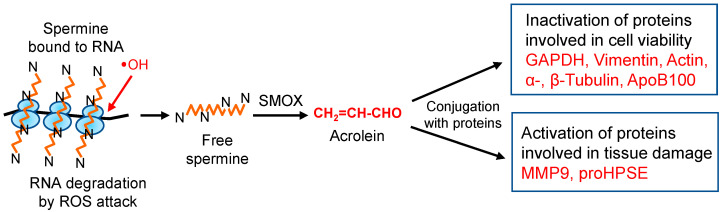
Change of protein activities through acrolein conjugation. In eukaryotic cells, 85% of spermine exists as an RNA–spermine complex in a non-covalent form [63]. When ribosomes are attacked by hydroxyl radicals, one of the ROS, spermine, can be released from ribosomes in a free form. Then, spermine is mainly oxidized to acrolein via 3-aminopropanal by spermine oxidase (SMOX). Produced acrolein modifies proteins, either by activation or by inactivation, resulting in aggravation of diseases.

## Data Availability

Not applicable.

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
