# Peer review of "Molecular Characteristics of Toxicity of Acrolein Produced from Spermine"

_biomolecules, 2023, doi:10.3390/biom13020298_

Round 1

Reviewer 1 Report

The current manuscript by Kashiwagi and Igarashi represents a comprehensive review of the pathological consequence of acrolein-conjugated proteins.  Acrolein is a highly reactive aldehyde primarily produced from the polyamine spermine after its oxidation by the catalolic enzyme, spermine oxidase.  By covering a variety of pathological states associated with acrolein-conjugated proteins, the authors provide the reader with a useful review of the topic. 

Although this review is generally well-written and comprehensive there are a few issues that should be addressed prior to publication.

1)  On page 2, the description of the acrolein-resistant Neuro2a cells is somewhat clumsy.  The use of the term “acrolein toxicity decreasing” is not readily understood.  It is suggested that the authors use the term “acrolein-resistant” instead to more clearly describe the cell lines that they have developed.

2) The accepted, official gene/enzyme name abbreviations of the polyamine catabolic enzymes are: spermine oxidase, SMOX; acetylpolyamine oxidase, PAOX; spermidine/spermine N1-acetyltransferase, SAT1.  To maintain consistency with the current literature it would be best if these abbreviations were used.

3) It is true that acrolein is more acutely toxic that most ROS, this is especially true with respect to protein damage.  However, it would be appropriate for the authors to indicate in their conclusions that ROS damage can be associated with gene and epigenetic changes that are less likely to occur with acrolein and can lead to other pathologies such as cancer.

Author Response

Dear Reviewer 1

We appreciate your helpful suggestions.

We combined Figures 3 and 4, and added new Figure 8 containing summary.

We removed some parts of Figure 1, Figure 2 and Figure 4 (old Figure 5).

Responses

1) On page 2, the description of the acrolein-resistant Neuro2a cells is somewhat clumsy. The use of the term “acrolein toxicity decreasing” is not readily understood. It is suggested that the authors use the term “acrolein-resistant” instead to more clearly describe the cell lines that they have developed.

Response:

In original report, the words “acrolein toxicity decreasing” were used.  So, on page 2, we changed from “acrolein toxicity decreasing” to “acrolein toxicity decreasing, i.e., acrolein-resistant”

2) The accepted, official gene/enzyme name abbreviations of the polyamine catabolic enzymes are: spermine oxidase, SMOX; acetylpolyamine oxidase, PAOX; spermidine/spermine N1-acetyltransferase, SAT1. To maintain consistency with the current literature it would be best if these abbreviations were used.

Response:

We changed the abbreviations as you suggested.

3) It is true that acrolein is more acutely toxic that most ROS, this is especially true with respect to protein damage. However, it would be appropriate for the authors to indicate in their conclusions that ROS damage can be associated with gene and epigenetic changes that are less likely to occur with acrolein and can lead to other pathologies such as cancer.

Response:

We described in “Concluding remarks”.

We hope that our revised review article is suitable for publication in Biomolecules.

With best wishes,

Keiko Kashiwagi

Reviewer 2 Report

Biogenic polyamines spermidine and spermine are essential, ubiquitous, and organic polycations present in all eukaryotic cells in µM–mM concentrations and are involved in the regulation of numerous vital processes required for the differentiation, proliferation, etc. Disturbance of polyamine metabolism is associated with many diseases, including malignant tumors (cancer cells have elevated level of polyamines), decreased immune response, some types of pancreatitis, Snyder–Robinson’s syndrome, and Alzheimer’s and Parkinson’s disease. The current review by Kashiwagi and Igarashi is devoted to a hot topic – the molecular characteristics of the toxicity of acrolein produced from spermine.

Spermine is catabolized in two different ways, and both leads to spermidine and toxic hydrogen peroxide and acrolein. First is the direct oxidation by SMO and second is a two step process – acetylation by SSAT and subsequent degradation by APAO. Igarashi and others have reported that it is just Spm, but not Spd or unsaturated fatty acids, is the source of acrolein in Neuro2a cells and shown that acrolein is detoxified by glutathione.

The next part of the review describes the results of different articles dedicated to GAPDH (glyceraldehyde-3-phosphate dehydrogenase) and it’s conjugates with acrolein. It was shown that acrolein same as nitric oxide (NO·) reacts with cysteine residue of GAPDH, then acrolein-conjugated GAPDH is translocated to nuclei and as a result causes apoptosis.

Besides Igarashi and colleagues has identified that acrolein conjugates not only with vimentin but also with actin and α- and β-tubulin proteins via reactive Cys- and Lys-residues, using Neuro2a cells. As a result, dendritic spine extension is significantly reduced, and it caused functional defects in brain signaling.

Authors also reviewed investigations devoted to the activation of metalloproteinase (MMP-9) by acrolein, the effects of acrolein conjugation to immunoglobulin, and other milestone data related to medicinal biochemistry of acrolein.

In general, the review is of significant interest to wide scientific audience. The authors are among the founders of these studies and in the present review critically discuss and highlight hot papers of their and other research groups involved in the study of acrolein toxicity and functions.

Author Response

Dear Reviewer 2

We appreciate your warm comments.

We combined Figures 3 and 4, and added new Figure 8 containing summary.

We removed some parts of Figure 1, Figure 2 and Figure 4 (old Figure 5).

We hope that our revised review article is suitable for publication in Biomolecules.

With best wishes,

Keiko Kashiwagi

Reviewer 3 Report

Regrettably, I am unwilling to complete a scientific review of this paper while it contains material published elsewhere.

Author Response

Dear Reviewer 3

I am sorry to bother you.

We combined Figures 3 and 4, and added new Figure 8 containing summary.

We removed some parts of Figure 1, Figure 2 and Figure 4 (old Figure 5).

We hope that our revised review article is suitable for publication in Biomolecules.

With best wishes,

Keiko Kashiwagi